# Heterogeneous Transcriptional Landscapes in Human Sporadic Parathyroid Gland Tumors

**DOI:** 10.3390/ijms251910782

**Published:** 2024-10-07

**Authors:** Chiara Verdelli, Silvia Carrara, Riccardo Maggiore, Paolo Dalino Ciaramella, Sabrina Corbetta

**Affiliations:** 1Laboratory of Experimental Biochemistry and Molecular Biology, IRCCS Ospedale Galeazzi-Sant’Ambrogio, 20157 Milan, Italy; chiara.verdelli@grupposandonato.it; 2Department of Medical Biotechnology and Translational Medicine, University of Milan, 20133 Milan, Italy; silvia.carrara1@unimi.it; 3Endocrine Surgery, IRCCS Ospedale San Raffaele, 20132 Milan, Italy; maggiore.riccardo@hsr.it; 4Endocrine Unit, ASST Ospedale Niguarda, 20162 Milan, Italy; pdalino@icloud.com; 5Bone Metabolism Diseases and Diabetes Unit, IRCCS Istituto Auxologico Italiano, 20145 Milan, Italy; 6Department of Biomedical, Surgical and Dental Sciences, University of Milan, 20122 Milan, Italy

**Keywords:** parathyroid tumors, MEN1, CDC73, calcium-sensing receptor, parathormone, microRNAs, lncRNAs

## Abstract

The expression of several key molecules is altered in parathyroid tumors due to gene mutations, the loss of heterozygosity, and aberrant gene promoter methylation. A set of genes involved in parathyroid tumorigenesis has been investigated in sporadic parathyroid adenomas (PAds). Thirty-two fresh PAd tissue samples surgically removed from patients with primary hyperparathyroidism (PHPT) were collected and profiled for gene, microRNA, and lncRNA expression (n = 27). Based on a gene set including *MEN1*, *CDC73*, *GCM2*, *CASR*, *VDR*, *CCND1*, and *CDKN1B*, the transcriptomic profiles were analyzed using a cluster analysis. The expression levels of *CDC73* and *CDKN1B* were the main drivers for clusterization. The samples were separated into two main clusters, C1 and C2, with the latter including two subgroups of five PAds (C2A) and nineteen PAds (C2B), both differing from C1 in terms of their lower expression of *CDC73* and *CDKN1B*. The C2A PAd profile was also associated with the loss of TP73, an increased expression of *HAR1B*, *HOXA-AS2*, and *HOXA-AS3* lncRNAs, and a trend towards more severe PHPT compared to C1 and C2B PAds. C2B PAds were characterized by a general downregulated gene expression. Moreover, *CCND1* levels were also reduced as well as the expression of the lncRNAs *NEAT1* and *VLDLR-AS1*. Of note, the deregulated lncRNAs are predicted to interact with the histones H3K4 and H3K27. Patients harboring C2B PAds had lower ionized and total serum calcium levels, lower PTH levels, and smaller tumor sizes than patients harboring C2A PAds. In conclusion, PAds display heterogeneous transcriptomic profiles which may contribute to the modulation of clinical and biochemical features. The general downregulated gene expression, characterizing a subgroup of PAds, suggests the tumor cells behave as quiescent resting cells, while the severity of PHPT may be associated with the loss of p73 and the lncRNA-mediated deregulation of histones.

## 1. Introduction

Primary hyperparathyroidism (PHPT) is a common metabolic disorder sustained by tumors of the parathyroid glands which release excessive amounts of parathyroid hormone (PTH). Inappropriate PTH secretion leads to hypercalcemia and has consequences mainly on patients’ bones and kidneys. Most parathyroid tumors are benign sporadic adenomas (PAds). The aberrant expression of several genes has been demonstrated to be involved in parathyroid tumorigenesis. Germline mutations of genes, such as *MEN1*, *CDC73*, *CDKN1B*/p27, *GCM2*, and *CASR*, generate hereditary syndromes, namely, multiple neoplasia syndrome type 1 (OMIM#131100), primary hyperparathyroidism-jaw tumors (JT-HPT, OMIM#145000), multiple neoplasia syndrome type 4 (OMIM#610755), familial isolated hyperparathyroidism 4 (HRPT4; OMIM#617343), familial hypocalciuric hypercalcemia (HHC1; OMIM#145980), and neonatal severe hyperparathyroidism (NSHPT; OMIM#239200).

Mutated genes involved in these hereditary syndromes include oncosuppressors, embryonic transcription factors, cell cycle regulators, and G-protein coupled receptors. Each gene has been demonstrated to cause parathyroid neoplasia. Men1 conditional heterozygous mice, whose *Men1* exons 1 and 2 were deleted to resemble MEN1 syndrome, developed parathyroid hyperplasia by 9 months of age, and Men1 conventional mice developed parathyroid adenomas by 15 months [1]. A loss of heterozygosity and menin expression were demonstrated in the parathyroid tumors. Men1 heterozygotic mice with parathyroid neoplasms were hypercalcaemic and hypophosphataemic, with inappropriately normal serum parathyroid hormone concentrations [2]. Mice with a parathyroid-specific deletion of Cdc73, the gene involved in JT-HPT syndrome, developed parathyroid tumors from 9 months of age, which showed nuclear pleomorphism, fibrous septation, and increased galectin-3 expression, consistent with atypical parathyroid adenomas associated with PHPT [3]. Variants of *GCM2*, a parathyroid-specific embryonic transcription factor, increase the transactivation of the PTH promoter in vitro and are associated with familial isolated hyperparathyroidism (FIHP; OMIM#614373 HRPT4) [4]. In vitro activating *GCM2* variant alleles are significantly overrepresented in PHPT patients with multiglandular or familial disease compared to the general population, yet penetrance values are very low [5,6,7,8]. Among cell cycle regulators, the inactivation of the *Cdkn1b*/p27 gene in mice and rats induces the development of a phenotype resembling the cell proliferations observed in multiple endocrine glands in humans, including parathyroid adenomas [9,10]. Moreover, the overexpression of the cyclin D1 (*CCND1*) oncogene in parathyroid cells results in parathyroid hypercellularity with a phenotype of chronic biochemical PHPT and an abnormal in vivo PTH-calcium set point; in this PHPT model, abnormal parathyroid proliferation regularly preceded the dysregulation of the calcium-PTH axis [11]. Finally, ablating *CASR*, which encodes for the calcium-sensing receptor, in the parathyroid glands increases PTH synthesis and parathyroid gland growth [12]. Casr heterozygous mice, analogous to humans with familial hypocalciuric hypercalcemia (HHC1), had benign and modest elevations of serum calcium, magnesium, and parathyroid hormone levels, as well as hypocalciuria. In contrast, Casr homozygous mice, like humans with NSHPT, had markedly elevated serum calcium and parathyroid hormone levels and parathyroid hyperplasia [13]. The deletion of the *VDR* specifically in the parathyroid glands decreases parathyroid calcium-sensing receptor expression with a moderate increase in basal PTH levels with normal serum calcium levels, while total-body *VDR* deletion is associated with an increase in the number of proliferating parathyroid cells [14].

Somatic mutations of the same genes have been found in sporadic PAds [15]. Moreover, epigenetic deregulation emerges in sporadic parathyroid tumors [16], in which several genes, coding for long noncoding RNAs, microRNAs, and circRNAs, have been found to be deregulated [17].

Therefore, the complexity of the genetic and epigenetic landscape of parathyroid tumorigenesis is emerging. Sporadic PAds are the most common parathyroid neoplasia as they are detected in about 60% of PHPT patients. Among sporadic PAds, the contribution of each genetic and/or epigenetic aberration and their interactions is poorly understood. Indeed, analysis of genetic and epigenetic aberrations may contribute to better define PAds’ clinical and hormonal behavior and to move towards personalized treatment.

By simultaneously analyzing the expression of a set of parathyroid-related genes in a series of sporadic parathyroid tumors derived from patients with PHPT, this study aimed to identify different genetic and epigenetic profiles correlating with clinical and biochemical PHPT features.

## 2. Results

### 2.1. Clinical Features of PHPT Patients Harbouring PAds

Thirty-two patients (six males and 26 females, aged 58.1 ± 10.5 years, mean ± SD) affected with PHPT and harboring a single parathyroid gland tumor (PAd) were enrolled. Their ionized calcium levels (mean ± SD) were 1.53 ± 0.52 mmol/L, their serum total calcium levels were 12.0 ± 2.3 mg/dL, and their plasma PTH levels were 256.0 ± 330.6 pg/mL. All patients had serum creatinine levels and an estimated glomerular filtration rate (eGFR) in the normal range. Their urine calcium excretion levels (mean over 24 h) were 412.4 ± 138.2 mg/24 h. Symptomatic or asymptomatic kidney stones were detected in 46.9% of the PHPT patients, while osteoporosis was diagnosed in 62.5% of the patients. Hypertension was reported by 31.2% of the patients.

### 2.2. Gene Expression by Real-Time PCR

Genes encoding for the oncosuppressors *MEN1*, *YAP1*, *CTNNB1*, *RASSF1A*, and *CDC73* and the parathyroid embryonic transcription factors *TBX1*, *PAX1*, *GCM2*, and *GATA3* were expressed in all of the PAd samples. Similarly, all of the PAd samples showed the expression of the parathyroid-specific genes *PTH*, *CASR*, *VDR,* and *GPRC6A*. Regarding cell cycle genes, it is of note that two PAd samples significantly overexpressed *CCND1*. *CDKN1B*/p27 and *CDKN1A*/p21 were variably expressed. In contrast, *TP73* gene expression was undetectable in 13 (40.6%) out of 32 PAd samples. miR-517c and miR-372-3p, belonging to the two close microRNA clusters C19MC and miR-371-373 on chromosome 19, were concomitantly overexpressed in four (12.5%) out of the thirty-two PAd samples, while miR-126-3p and miR-93-5p were variably expressed in all of the samples. *NEAT1* transcripts were expressed in all of the PAd samples, while *HAR1B* could not be detected in four (12.5%) PAd samples, *HOXA-AS2* in six samples (19.0%), and *HOXA-AS3* in twenty samples (62.5%). The long non-coding RNAs *VLDLR-AS1* and *SHNG6* were variably expressed in all of the samples.

### 2.3. Unsupervised Clustering Analysis

The whole gene expression set was analyzed considering the expression of genes whose encoded proteins were previously demonstrated to be involved in parathyroid pathophysiology with consistent experimental evidence. Seven genes, including *MEN1*, *CDC73*, *GCM2*, *CASR*, *VDR*, *CCND1*, and *CDKN1B*, were selected.

PAd32 overexpressing *CCND1* was separately clustered from the remaining PAds, and it was assumed to belong to the nearest cluster (Figure 1). The samples were clustered in two main clusters, named Cluster 1 (C1), including eight PAd samples, and Cluster 2 (C2), including five PAd samples (C2A) distinct from the remaining 19 PAds (C2B) (Figure 1).

A principal component analysis identified low expression levels of *CCND1* as the main component of variance (PC1, 61.7%), the second component was characterized by low expression levels of *CCND1* and *MEN1* (PC2, 15.2%), and the third component included low levels of *MEN1* and *GCM2* (PC3, 9.6%).

### 2.4. Selected mRNA, miRNA, and lncRNA Expression Profiles in the Different PAd Clusters

PAds included in both C2A and C2B showed significant downregulation in the expression of *CDC73* and *CDKN1B* compared with PAds in C1 (Table 1). PAds in C2B, but not PAds in C2A, were expressed at significantly lower levels, similarly to *MEN1*, *GCM2*, *CASR*, *VDR*, and *CCND1*; indeed, these genes showed a trend towards downregulated expression in the PAds of C2A, though this reduction was not statistically significant, suggesting that C2A may represent an intermediate gene profile between C1 and C2B (Table 1). The downregulated expression of the genes considered for clusterization was associated with the reduced expression of *RASSF1A*, *YAP1*, *CTNNB1*, *GATA3*, and *PTH*. Of note, *TP73* was expressed in all C1 samples, while the gene was undetectable in 80% of C2A and 47.4% of C2B samples (C1 vs. C2B, *p* = 0.0109).

Though microRNA expression did not significantly differ among the three clusters, the four PAds overexpressing miR-372-5p and miR-517c, belonging to the C19MC cluster, were included in clusters 2A and 2B. Moreover, long non-coding RNA expression significantly varied among the clusters. *NEAT1* and *VLDLR-AS1* expression levels were downregulated in C2B PAds, while PAds in C2A expressed *HOXA-AS2* and *HOXA-AS3* at higher levels and *HAR1B* at lower levels than those detected in C1 and C2B PAds.

### 2.5. Clinical Features Associated with the Different Gene Profiles

The PAds in the three clusters were from PHPT patients similar in age, sex distribution, and PHPT-related complications, namely, the prevalence of kidney stones, osteoporosis, and arterial blood hypertension (Table 1). Of note, patients harboring C2B PAds had ionized calcium, total serum calcium, and PTH levels significantly lower than those of patients harboring C2A PAds, also showing a trend towards smaller tumor sizes (Table 2). Kidney function, evaluated by the estimated glomerular filtration rate, was similar among the three clusters.

### 2.6. Prediction of Intracellular Pathway Deregulated in PAds

Considering the deregulated genetic profiles derived from the cluster analysis, we analyzed the interactions among the proteins encoded by the selected genes using the STRING database (https://string-db.org; version 12.0, accessed on 15 September 2024). The STRING analysis revealed a narrow interacting network of the selected genes (Appendix A), in which the Markov Cluster (MCL) algorithm identified two main domains: (1) the β-catenin-TCF complex and WNT signaling regulated by RUNX3, and (2) parathyroid hormone synthesis, secretion, and action (Appendix A). The MCL is an unsupervised cluster algorithm for graphs/networks based on simulations of (stochastic) flow in graphs.

We also investigated interconnections by using the NDEx Biological Network Repository (Interactome, Signor Complete-Human, accessed on 30 July 2024) [19] to identify the pathways potentially involved. Genes deregulated in the C2A profile were involved in three network nodes, with one edge, while genes deregulated in the C2B profile were involved in 14 nodes and 11 edges. The networks indicated that the proteins encoded by the selected genes are connected by activation/inactivation relationships that have been described in the literature and annotated in SIGNOR [20]. The connections identified in the C2B PAd pathways are involved in cell proliferation, the G0 to G1 phase transition, and adipogenesis (Appendix A), while C2A PAds showed the deregulation of non-interconnected genes, though both were involved in the regulation of cell proliferation and cell cycle progress/block (Appendix A). Indeed, these tools did not include microRNAs or lncRNAs.

Considering the lncRNAs differentially expressed among the clusters, NEAT1, HAR1B, HOXA-AS2, HOXA-AS3, and VLDLR-AS were predicted by the RNA Interactome Database (http://rnainter.org) to target genes encoding for histones, namely H3K4me3 and H3K4me1, H3K27me3, and H3K27ac.

## 3. Discussion

The results of the present study suggest that sporadic PAds emerge as benign tumors with heterogeneous transcriptomic profiles. At variance with previous studies extensively investigating gene expression by throughput approaches comparing parathyroid adenomas with carcinomas or, less frequently, with normal parathyroid tissue, here, we focused our attention on sporadic PAds, which have been considered up to now as a unique entity. Indeed, sporadic PAds are associated with a wide spectrum of clinical presentations spanning from asymptomatic mild hypercalcemia to life-threatening hypercalcemic crises [21,22]. PAds also present several somatic gene mutations. It is conceivable that the aberration of multiple molecular pathways may be involved in determining different clinical phenotypes of sporadic PAds.

Here, we limited the transcriptomic analysis to a set of 27 genes previously suggested to be involved in parathyroid tumorigenesis to evaluate their contribution; approaching the analysis by throughput methods may reduce the sensitivity of the analysis and mild changes in gene expression may be undetectable. This item may be relevant considering that previous studies found general gene downregulation in PAds [23]. Moreover, transcriptomic analysis has been extended to microRNAs and lncRNAs relevant in parathyroid tumors to provide insight about non-coding-RNA-mediated epigenetic modulation.

Two main distinct transcript profiles were identified, substantially differing in the general downregulated expression of the seven genes selected for the clustering, which is in line with recently published data obtained by whole-exome sequencing in a series of 41 sporadic PAds, in which most genes were downregulated [23]. The genes considered for the clusterization highlighted the involvement of the β-catenin-RUNX3 pathway in addition to the parathyroid-specific pathway regulating calcium sensitivity and PTH synthesis and secretion. Of note, the β-catenin-RUNX3 pathway is a complex regulatory network involved in tumorigenesis [24].

The diversity of the expression levels of *CDC73* and *CDKN1B*/p27 was the main driver of the clusterization. These were downregulated in PAds included in both C2 clusters compared with PAds included in C1. Indeed, the C2A PAds presented intermediate transcript profiles between those detected in C1 PAds and C2B PAds, in which *CDC73* and *CDKN1B* downregulation was associated with the substantially conserved expression levels of all other analyzed protein-coding genes. Furthermore, most C2A PAds had decreased *TP73* expression. TP73 is a member of the p53 family, exerting many activities spanning from embryonic development through tumor suppression; it is believed to likely play a dual role as a tumor suppressor by regulating apoptosis in response to genotoxic stress or as an oncoprotein by promoting the immunosuppressive environment and immune cell differentiation [25]. The NDEx interactome tool suggested that the C2A transcript profiles may involve pathways regulating cell cycle progression/block.

The intermediate profile of the C2A PAds was associated with the overexpression of specific lncRNAs, namely *HAR1B*, *HOXA-AS2,* and *HOXA-AS3*. HOXA-AS3 exhibited significant properties in regulating several biological processes in different human cancers, including cell proliferation, invasion, and migration [26]. Similarly, the high level of expression of *HOXA-AS2* is associated with a poor prognosis and the clinicopathological characteristics of cancer patients [27]. The precise role of *HAR1B* still needs to be elucidated; nonetheless, it has been found to be upregulated in cancers [28] and likely induces *SOX2* expression in PAd-derived cells [29].

Therefore, C2A PAds, which represented a small proportion of the analyzed PAd series, emerged as tumors with a reduced expression of the oncosuppressors *CDC73* and *CDKN1B*, the loss of *TP73,* and an increased expression of the lncRNAs *HAR1B*, *HOXA-AS2*, and *HOXA-AS3*, sharing features with most common human cancers; this transcript profile was associated with a trend towards more severe PHPT. Considering that the downregulated expression of the oncosuppressors and the loss of p73 is shared with PAds included in the C2B cluster, we are tempted to speculate that more severe PHPT features may be associated with the increased expression of lncRNAS, which are predicted to interact with H3K4 and H3K27 histones.

In contrast, the C2B PAds differed as they displayed a significant downregulation of the oncosuppressors *MEN1*, *RASSF1A*, *YAP1*, and *CTNNB1*. The C2B PAds showed low transcript levels of the parathyroid-specific embryonic transcription factors *GCM2* and *GATA3*. Besides the loss of oncosuppressors and differentiation-promoting transcription factors, C2B PAds displayed the cell cycle regulatory genes *CCND1* and *CDKN1B* which were definitely downregulated. Moreover, C2B PAds had low *CASR* and *VDR* expression, and unexpectedly low *PTH* transcript levels. Indeed, the reduced expression of *CASR* is in line with low expression levels of *GCM2*, which has been demonstrated to maintain high levels of CASR expression in parathyroid cells [30]. An investigation using NDEX tools (https://www.ndexbio.org/index.html#/; accessed on 30 July 2024) [19] of the intracellular signaling pathways potentially deregulated by the aberrant transcript profile in C2B PAds suggested the involvement of regulators of the G0 to G1 phase transition, cell proliferation, and adipogenesis, which may be inhibited. The downregulation in C2B PAds involved also lncRNAs; in particular, C2B PAds had reduced expression levels of *NEAT1* and *VLDLR-AS1*. Nuclear paraspeckle assembly transcript 1 (NEAT1) is elevated in several types of cancer and promotes cancer growth. Of note, recent studies have also demonstrated that the knockdown of NEAT1 can inhibit cancer cells’ proliferation, movement, and infiltration while enhancing apoptosis [31]. Downregulated NEAT1-mediated promotion of apoptosis is in line with the mild phenotype detected in C2B PAds, which were smaller and associated with less severe PHPT. The role of VLDLR-AS1 is largely undefined, though it has been reported to be upregulated in thymomas compared with normal thymuses [32]. Considered as a whole, the lncRNAs deregulated among the Pad clusters are predicted to modulate histones, namely H3K4 and H3K27, which have been reported to critically interact with crucial proteins involved in parathyroid tumorigenesis such as menin and parafibromin [33].

As far as microRNA is concerned, three out of four PAds overexpressing miR-372-5p and miR-517c, members of the embryonic microRNA clusters located on chromosome 19, were included in C2B; this finding was unexpected as the aberrant expression of C19MC has been associated with parathyroid carcinomas and atypical parathyroid adenomas [34].

Finally, PAds included in C2B with the most downregulated genes presented a significantly smaller tumor size; this finding is unexpected as the loss of oncosuppressors and cyclin inhibitors has been associated with enhanced cell proliferation in most cancer models. Moreover, this resembles the features of quiescent resting cells.

The present study based on the investigation of a specific set of genes is innovative and provides perspectives for the targeted management of PHPT. Admittedly, the study suffered from some limits: (1) the set of explored genes is limited though it included genes with a relevant role in parathyroid tumorigenesis; (2) the set of analyzed PAd samples is small though homogeneous; (3) the analysis is limited to a unique technical approach; (4) the transcript data could not be confirmed by the analysis of the corresponding protein levels; and (5) the PAd gene expression profiles were not compared with those in normal parathyroid glands due to their unavailability for ethical reasons. Though the lack of normal parathyroid samples may limit the significance of the present study in investigating the pathogenesis of parathyroid tumors, we explored the genetic and epigenetic heterogeneity among the benign parathyroid adenomas considering the wide spectrum of clinical presentations associated with parathyroid tumors.

## 4. Materials and Methods

### 4.1. Parathyroid Tumor Samples

Fresh adenomatous parathyroid tissue samples (PAds, n = 32) were collected from PHPT patients undergoing to parathyroid surgery and used for gene, miRNA, and lncRNA profiling. PAd samples were collected immediately after surgery and snap-frozen. The diagnosis of PHPT and histologic classification of PAds were made according to the most recent guidelines [19,35]. Histopathological examination was confirmed by two independent pathologists according to 2022 WHO criteria [19].

The study was approved by an Institutional Ethical Committee (Ospedale San Raffaele Ethical Committee, protocol no. GPRC6A PARA, 07/03/2019; CE40/2019), and informed consent was obtained from all patients. Fasting plasma ionized calcium, serum total calcium, and PTH were routinely measured to diagnose PHPT.

### 4.2. RNA Extraction

RNA extraction from frozen tissue was performed in 1 mL of Trizol reagent (Invitrogen) following the manufacturer’s instructions using a tissue homogenizer. RNA was quantified by spectrophotometry at 260 nm and DNA contamination was removed by DNase I (Life Technologies, Carlsbad, CA, USA). cDNA for the gene expression was obtained using iScript cDNA Synthesis Kit (BIORAD) starting from total RNA of 300 ng. TaqMan microRNA reverse transcription kit (Applied Biosystems, Waltham, MA, USA) was applied to synthesize the cDNA chain from miRNA with specific RT primer.

### 4.3. Real-Time PCR

Genes encoding the oncosuppressors *MEN1* and *CDC73*, the parathyroid embryonic transcription factor *GCM2*, the cell cycle genes *CCND1* and *CDKN1B*, and the parathyroid membrane receptors *CASR* and *VDR* were analyzed, considering that their direct involvement in parathyroid tumorigenesis has been experimentally demonstrated for each gene (Appendix A). The transcript profiling of PAds was extended to further 20 genes, including 3 oncosuppressors (*RASSF1A*, *YAP1*, and *CTNNB1*), 3 embryonic transcription factors (*TBX1*, *PAX1*, and *GATA3*), 2 parathyroid-specific genes (*PTH* and *GPRC6A*), 2 cell cycle regulators (CDKN1A and TP73), 4 microRNAs (miR-372-5p, miR-517c, miR-126-3p, and miR-93-5p), and 6 lncRNAs (NEAT1, HAR1B, HOXA-AS2, HOXA-AS3, VLDLR-AS1, and SNHG6) (Appendix A). All genes have been reported to be deregulated in PAds in previous studies.

Real-time PCR was performed using QuantStudio™ 12K Flex System (Applied Biosystems), and the following gene expression assays (all from Thermo Fisher Scientific) were used: MEN1 (Hs00365720_m1), YAP1 (Hs00371735_m1), RASSF1A (Hs00200394_m1), CDC73 (Hs00363810_m1), PTH (Hs00757710_g1), TBX1 (Hs00271949_m1), PAX1 (Hs01071292_m1), GCM2 (Hs00899403_m1), GATA3 (Hs00231122_m1), CASR (Hs01047795_m1), VDR (Hs01045843_m1), GPRC6A (Hs01026851_m1), CCND1 (Hs00765553_m1), CDKN1B (Hs01597588_m1), CDKN1A (Hs00355782_m1), CTNNB1 (Hs00355045), TP73 (Hs01056231_m1) HAR1B (Hs03299152_m1), HOXA3as (Hs00940777_m1), HOXA6as (Hs03296494_m1), NEAT1 (Hs01008264_s1), SNHG6 (Hs00417251_m1), and lincRNA-VLDLR (Hs03309900_m1). Gene expression levels were analyzed using global mean normalization. For the amplification of miRNA, the following probes were used: miR-517c-3p (001153), miR-126-3p (002228), miR-93-5p (001090), miR-372-3p (000560), and RNU48 (001006). The miRNA expression levels were normalized to the RNU48 levels. Relative expression levels were calculated as 2−ΔCt.

### 4.4. Statistical Analysis

Testing the hypothesis that the set of 7 genes selected for the clustering analysis would significantly identify different transcriptomic profiles, using linear multiple regression test with two tails and designating an α error of 0.05 and an effect size of 0.33, the sample size of 32 PAds has a power (1-b error) of 0.88; calculation was performed by G*power 3.1.

Unsupervised hierarchical clustering analysis by Euclidean wardd2 was performed using Past4.0 to identify specific profiles. Principal component analysis (PCA) based on representation of correlations was developed to confirm the clustering.

Parametric ordinary one-way or non-parametric Kruskal–Wallis ANOVA adjusted for multiple testing were performed to analyze the differences among the 3 gene profile-based clusters as appropriated. Normality of the data in each cluster was tested by D’Agostino and Pearson omnibus normality test. Normally distributed data are presented as mean ± standard deviation, while data failing the normality test are presented as median and range interquartile. Differences among proportions were tested by Chi-square test. A *p* value of <0.05 was considered as statistically significant. Statistical analysis was performed using Prism 6.0.

## 5. Conclusions

The transcriptional landscape of parathyroid benign tumorigenesis is complex and far from that described in common human malignancies. The expression levels of *CDC73* and *CDKN1B*/p27, interacting with molecules involved in WNT/β-catenin and parathyroid-specific pathways and modulated by epigenetic mechanisms like lncRNAs and histone-related methylation, may determine the heterogeneity of the clinical features of parathyroid tumors. A network of genes has emerged whose variable expression may modulate clinical and biochemical features. It would be of interest to identify extracellular signaling molecules and/or pharmacological agents modulating gene expression in PAds. Therefore, the present data support the integration of genetic, epigenetic, and transcriptional aberrations, which can contribute to better define PAds’ clinical and hormonal tumoral behavior, moving toward personalized treatment and follow-up.

## Figures and Tables

**Figure 1 ijms-25-10782-f001:**
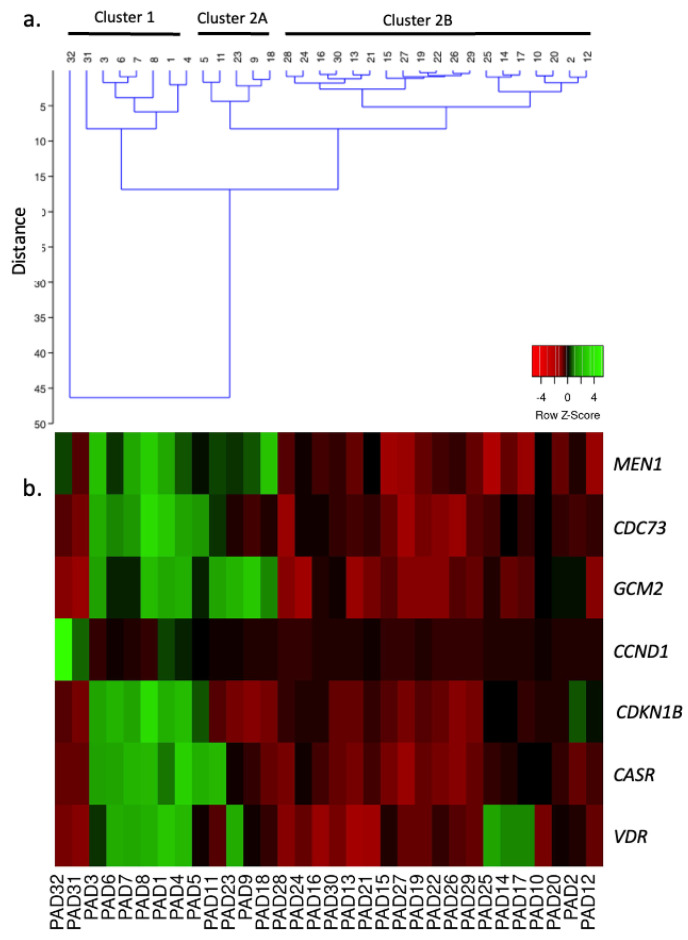
Hierarchical cluster analysis of analyzed PAd series based on the expression of the genes known to be pathogenic in parathyroid tumorigenesis. (**a**) Dendrogram identifying 3 clusters; distance, also known as dissimilarity, is represented on the vertical scale. (**b**) Heatmap representing the expression levels of each analyzed gene in each PAd sample.

**Table 1 ijms-25-10782-t001:** Transcript profiles of the clusters.

Gene	Cluster 1	Cluster 2A	Cluster 2B	*p*
*Oncosuppressors*				
*MEN1*	1.921 ± 0.718	1.706 ± 0.555	0.770 ± 0.290 *^,^**	**<0.0001**
*CDC73*	2.034 ± 1.046	1.252 ± 0.441 *	0.806 ± 0.269 *	**0.0001**
*RASSF1A*	1.417 (0.788, 2.022)	1.596 (1.155, 2.578)	0.836 (0.542, 1.048) **	**0.0439**
*YAP1*	2.033 ± 1.210	1.403 ± 0.450	0.788 ± 0.324 *	**0.0004**
*CTNNB1*	2.198 ± 1.290	1.511 ± 1.028	0.877 ± 0.421 *	**0.0021**
*Transcription factors*				
*GCM2*	1.693 ± 0.955	2.315 ± 0.713	0.713 ± 0.357 *	**<0.0001**
*GATA3*	1.629 (1.260, 2.498)	1.202 (0.819, 1.462)	0.857 (0.580, 1.040) *	**0.0041**
*TBX1*	1.157 (0.704, 2.010)	1.711 (1.027, 2.211)	0.806 (0.540, 1.091)	0.1292
*PAX1*	2.058 (0.854, 2.884)	0.803 (0.533, 2.012)	1.089 (0.515, 1.308)	0.0742
*Parathyroid-specific genes*				
*PTH*	2.577 (1.393, 7.689)	1.218 (0.488, 3.108)	0.811 (0.509, 1.123) *	**0.0052**
*CASR*	2.380 ± 1.252	1.833 ± 1.252	0.756 ± 0.313 *^,^**	**0.0001**
*VDR*	2.972 (0.868, 3.417)	1.298 (0.915, 2.201)	0.713 (0.498, 1.206) *	**0.0252**
*GPRC6A*	0.999 (0.542, 2.569)	2.092 (0.931, 4.779)	0.983 (0.364, 2.262)	0.4753
*Cell cycle genes*				
*CCND1*	2.340 (0.685, 5.710)	1.484 (0.873, 2.008)	0.676 (0.502, 0.954) *^,^**	**0.0058**
*CDKN1B/p27*	3.723 ± 2.229	0.809 ± 0.982 *	1.039 ± 0.634*	**<0.0001**
*CDKN1A/p21*	2.342 (1.186, 4.633)	1.227 (0.193, 2.820)	1.025 (0.292, 1.681)	0.0679
*TP73*	1.232 (0.579, 14.48)	0.213 (0.213, 6.900)	0.511 (0.213, 5.485)	0.0930
*MicroRNAs*				
miR-372-5p	0.268 (0.194, 2.463)	0.532 (0.304, 24.51)	0.353 (0.231, 0.927)	0.5641
miR-517c	0.217 (0.052, 0.752)	0.098 (0.061, 7674)	0.424 (0.135, 1.628)	0.2115
miR-126-3p	0.744 ± 0.504	0.951 ± 0.498	1.270 ± 0.502	0.0507
miR-93-5p	0.681 ± 0.423	1.173 ± 0.412	1.068 ± 0.396	0.0560
*Long non-coding RNAs*				
*NEAT1*	2.386 (1.636, 6.208)	2.285 (1.629, 3.635)	0.641 (0.402, 0.966) *^,^**	**<0.0001**
*HAR1B*	2.059 (0.519, 15.45)	0.130 (0.075, 0.698) *	2.507 (0.467, 3.592) **	**0.0371**
*HOXA-AS2*	0.348 (0.092, 1.871)	9.099 (3.443, 16.30) *	0.866 (0.079, 1.496) **	**0.0125**
*HOXA-AS3*	0.458 (0.458, 1.049)	4.784 (2.910, 11.92) *	0.458 (0.458, 1.302) **	**0.0028**
*VLDLR-AS1*	2.279 (1.278, 3.423)	1.976 (0.812, 2.556)	0.760 (0.303, 1.521) *	**0.0139**
*SNHG6*	1.681 (0.875, 2.607)	0.795 (0.509, 1.917)	0.971 (0.513, 1.383)	0.1202

*, *p* < 0.05 vs. Cluster 1; **, *p* < 0.05 vs. Cluster 2A; *p* was determined by one-way or Kruskall–Wallis ANOVA test adjusted for multiple comparisons when appropriate.

**Table 2 ijms-25-10782-t002:** Clinical and biochemical features of the PHPT patients harboring the analyzed PAds according to the different gene expression profiles.

Features	n.v.	Cluster 1	Cluster 2A	Cluster 2B	*p*
General and tumor features					
Female/male (%)	-	8/0 (100.0) *	2/3 (40.0)	15/4 (78.9)	**0.0388**
Age (years)	-	55.3 ± 7.8	59.8 ± 12.3	58.3 ± 11.2	0.2583
BMI (kg/m^2^)	-	23.7 ± 2.6	24.7 ± 1.8	25.6 ± 6.0	0.2863
Tumor size (cm)	-	1.88 ± 1.03	1.98 ± 0.38	1.27 ± 0.51	**0.0376**
Biochemical and hormonal parameters					
Ca^2+^ (mmol/L)	1.15–1.29	1.46 (1.35, 1.63)	1.62 (1.53, 2.33)	1.41 (1.38, 1.49) *	**0.0398**
S Total Ca (mg/dL)	8.4–10.4	11.9 ± 1.2	14.7 ± 4.5	11.3 ± 0.7 *	**0.0097**
S Phosphate (mg/dL)	3.5–5.0	2.7 ± 0.7	2.5 ± 0.6	2.1 ± 0.7	0.2479
PTH (pg/mL)	10.0–65.0	170.5 (110.5, 372.5)	301.0 (195.0, 1031.0)	135.0 (88.1, 173.8) *	**0.0427**
eGFR (mL/min)	>90.0	85.8 ± 18.6	85.7 ± 16.5	83.2 ± 13.0	0.9278
UCa (mg/24 h)	<300.0	430.4 ± 183.6	451.4 ± 65.4	360.6 ± 128.9	0.3214
PHPT-related complications					
Kidney stones (y/n, %)	-	4/4 (50.0)	3/2 (60.0)	8/11 (42.1)	0.7593
Osteoporosis (y/n, %)	-	6/2 (75.0)	4/1 (80.0)	10/9 (52.6)	0.3723
Hypertension (y/n, %)	-	4/4 (50.0)	1/4 (20.0)	5/14 (26.3)	0.4027

* *p* < 0.05 vs. Cluster 2A, *p* was determined by one-way or Kruskall–Wallis ANOVA test adjusted for multiple comparisons or Chi-square test when appropriate; n.v., normal values; BMI, body mass index; Ca^2+^, plasma ionized calcium; S Total Ca, serum total calcium; S Phosphate, serum phosphate; PTH, circulating parathormone; eGFR, estimated glomerular filtration rate according KDIGO [18]; UCa, urine calcium excretion; PHPT, primary hyperparathyroidism; y/n, yes/no; kidney stones, history of renal colics and stone expulsion, surgery, or asymptomatic kidney stones at imaging; osteoporosis, osteoporosis, or osteopenia are defined by the WHO.

## Data Availability

The datasets generated and analyzed during the current study are available at 10.5281/zenodo.11081163.

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
