# Peer review of "Heterogeneous Transcriptional Landscapes in Human Sporadic Parathyroid Gland Tumors"

_ijms, 2024, doi:10.3390/ijms251910782_

Round 1
Reviewer 1 Report
Comments and Suggestions for Authors
IJMS-3164752: “Heterogeneous transcriptional landscapes in human sporadic parathyroid gland tumors” authored by Chiara Verdelli.
General Comments:
The authors investigated the expression patterns of key genes in parathyroid tumors with comparing biochemical characteristics. The authors examined a set of genes involved in parathyroid tumorigenesis in the sporadic parathyroid adenomas (PAds). As a result, the expression levels of CDC73 and CDKN1B 22 were the main drivers for clusterization. It was also revealed that PAds display heterogeneous transcriptomic profiles that may contribute to modulate clinical features. The downregulated gene expression detected in a subgroup of PAds suggests a behavior of quiescent resting cells. These finding are interesting in the field genetic characteristics in the Pads; however, more clinical and pathological significance would be needed to consider genetic trends of PAds and rationale of the genetic changes.
Specific Comments:
1. First of all, the authors need to explain more detailed reasons why the 32 cases of PAds were divided into 3 clusters C1, C2 and C2B by the expression profiles of CDC73 and CDKN1B.
2. It would be necessary to investigate the functional significance of the gene groups such as onco-suppressors, transcription factors, parathyroid-specific genes, cell cycle genes, and others. For instance, is there any interrelationships between the gene sets? How about the differential interrelationships among the cluster groups?
3. In particular, the expression patterns of parathyroid specific-genes are important to discuss the endocrinological and oncologic features on PAds. The authors should analyze these interrelationship of gene sets and discuss the results.
4. In Table 1, the detailed clinical data related to PTH and hypercalcemia would be needed. For example, serum phosphate, Alp, bone-Alp levels, %TRP, %FEca, nephrogenous-cAMP, and bone density in the long bones would be added. Also, the location and number of PAds, the radiologic features such as mibi-scintigraphy, ultrasounds and doppler findings are informative to discuss the clinical and genetic characteristics.
5. Since the number of the examined PAd samples are limited, the lack of investigation on the normal parathyroid tissues or hyperplasia tissues would be critical to compare the gene profiles of PAds.
Author Response
General Comments:
The authors investigated the expression patterns of key genes in parathyroid tumors with comparing biochemical characteristics. The authors examined a set of genes involved in parathyroid tumorigenesis in the sporadic parathyroid adenomas (PAds). As a result, the expression levels of CDC73 and CDKN1B 22 were the main drivers for clusterization. It was also revealed that PAds display heterogeneous transcriptomic profiles that may contribute to modulate clinical features. The downregulated gene expression detected in a subgroup of PAds suggests a behavior of quiescent resting cells. These finding are interesting in the field genetic characteristics in the Pads; however, more clinical and pathological significance would be needed to consider genetic trends of PAds and rationale of the genetic changes.
Specific Comments:
- First of all, the authors need to explain more detailed reasons why the 32 cases of PAds were divided into 3 clusters C1, C2 and C2B by the expression profiles of CDC73 and CDKN1B.
We thank the Reviewer for the comment; we realized that the clusterization criteria had not been clearly stated in the manuscript. Indeed, cluster analysis was based on the expression profile of 7 genes including MEN1, CDC73, GCM2, CASR, VDR, CCND1, CDKN1B, as reported at page 3 line 124-131.
- It would be necessary to investigate the functional significance of the gene groups such as onco-suppressors, transcription factors, parathyroid-specific genes, cell cycle genes, and others. For instance, is there any interrelationships between the gene sets? How about the differential interrelationships among the cluster groups?
We thank the Reviewer for the interesting suggestion; we reviewed our data looking for potential interrelationships between the gene sets by using the STRING and RNA Interactome tools; results are reported in Supplementary Figure 1 and described at page 6 lines 185-192 and at page 7 lines 204-206, respectively.
Interrelationships among the clusters were explored by NDEx interactome and reported in the section Discussion.
- In particular, the expression patterns of parathyroid specific-genes are important to discuss the endocrinological and oncologic features on PAds. The authors should analyze these interrelationship of gene sets and discuss the results.
Again, the Reviewer’s suggestion has been met in the section Discussion.
- In Table 1, the detailed clinical data related to PTH and hypercalcemia would be needed. For example, serum phosphate, Alp, bone-Alp levels, %TRP, %FEca, nephrogenous-cAMP, and bone density in the long bones would be added. Also, the location and number of PAds, the radiologic features such as mibi-scintigraphy, ultrasounds and doppler findings are informative to discuss the clinical and genetic characteristics.
We agree with the reviewer that clinical and biochemical parameters would be of great interest in relationship with the gene expression patterns; however, most clinical data were not available, we could only add serum phosphate and urine calcium excretion levels in Table 1. Please, pay attention that Table 1 has been moved ahead according to the Reviewer 2’s suggestion and renamed as Table 2.
- Since the number of the examined PAd samples are limited, the lack of investigation on the normal parathyroid tissues or hyperplasia tissues would be critical to compare the gene profiles of PAds.
We meet the concern raised by the Reviewer; however, we were not able to provide adequate samples of normal parathyroid glands, which can be obtained only if accidentally removed (and recognized) during surgery for thyroid diseases in normocalcemic patients, whose signed consent to participate in the study has been collected. Though lack of normal parathyroid samples may limit the significance of the present study in investigating the pathogenesis of parathyroid tumors, we aimed to described heterogeneity among the benign parathyroid adenomas considering the wide spectrum of clinical presentation associated to parathyroid tumors. This item has been explicated at pages 8-9 line 296-302.
Reviewer 2 Report
Comments and Suggestions for Authors
Authors characterize the moelcular profile of 32 parathyroid adenoma tissues removed from patients with primary hyperparathryroidism.
It is not clear the choice of the clusterization . No analysis was compared to normal parathryoid tissue.
In the Results chaper 2.1 Clinical features are set as Table 1, without any comment and any previous reference to Cluster1, Cluster2A and Cluster2B.
The paragrah line 81-93 is not clear . The phrase "Among sporadis PAds is too long and very difficult to apprehend".
Please simplify and make ti clear
In the results section it should first be clearly explained and argumented the choice of the gene panel, and the choice of the clusterization.
The particularities of the three clusters are well underlined in the Discussion section, but until there it is difficult to unedrstant the choice of the clusterization.
Why there were no comparison with normal parathyroid tissue, neither with parathyrodi cancer.
Heterogeneity is interesting and expected, but which would be the pontetial perspectives of yous clusterization analysis?
Please detail.
Comments on the Quality of English LanguageSome phrases are difficult to understand
Author Response
Authors characterize the molecular profile of 32 parathyroid adenoma tissues removed from patients with primary hyperparathyroidism.
It is not clear the choice of the clusterization. No analysis was compared to normal parathryoid tissue.
We thank the Reviewer for the comment, we realized that clusterization criteria were not clearly stated. We expanded the description of the analysis at page 3 lines 124-131.
In the Results chapter 2.1 Clinical features are set as Table 1, without any comment and any previous reference to Cluster1, Cluster2A and Cluster2B.
We than the reviewer for the observation, it has been amended. Clinical and biochemical data of the whole PHPT patients have been added in paragraph 2.1. Table 1 has been moved ahead in the manuscript and renamed as Table 2.
The paragraph line 81-93 is not clear. The phrase "Among sporadic PAds” is too long and very difficult to apprehend. Please simplify and make it clear
Thank you for the comment; the paragraph has been rephrased at lines 89-94
In the results section it should first be clearly explained and argumented the choice of the gene panel, and the choice of the clusterization. The particularities of the three clusters are well underlined in the Discussion section, but until there it is difficult to understand the choice of the clusterization.
We thank the reviewer for the comment; we clarify the gene panel choice at page line 124-132.
Why there were no comparison with normal parathyroid tissue, neither with parathyroid cancer.
We recognized that this is an important item; however, we were not able to collect fresh tissue samples from normal parathyroid glands derived from normocalcemic patients surgically treated. Though lack of normal parathyroid samples may limit the significance of the present study in investigating the pathogenesis of parathyroid tumors, we aimed to described heterogeneity among the benign parathyroid adenomas considering the wide spectrum of clinical presentation associated to parathyroid tumors. For this reason, and due to the very limited number of parathyroid carcinomas enrolled at our Institutions in the study period, samples from parathyroid carcinomas had not been analyzed.
This item has been explicated at pages 8-9 line 296-301.
Heterogeneity is interesting and expected, but which would be the potential perspectives of your clusterization analysis? Please detail.
We thank the reviewer for the suggestion, we tried to provide some potential perspective in the section Conclusions.
Round 2
Reviewer 1 Report
Comments and Suggestions for Authors
The authors appropriately improved their manuscript. The clinical data on Table 2 is also suitable for the clinical readers. The referee would like to add the implication and discussion regarding the gene expression and clinical characteristics to the abstract and discussion sections.
Author Response
We thank the Reviewer once more for the solicitation to find out potential connections of the gene expression profiles with the clinical features of hyperparathyroidism associated to parathyroid adenomas.
We provided further speculations in the Abstract at page 1 lines 28-30 and lines 33-35, in the Discussion section at page 8 lines 259-263 and in the section Conclusions at page 10 lines 381-382.
Changes have been highlighted in red.

Reviewer 2 Report
Comments and Suggestions for Authors
Authors have improved their manuscript. They lcearly replied to all suggestions
Comments on the Quality of English LanguageAdequate
Author Response
My co-Authors and I sincerely thank the Reviewer.